# A Novel BiOCl Based Nanocomposite Membrane for Water Desalination

**DOI:** 10.3390/membranes12050505

**Published:** 2022-05-10

**Authors:** Rokhsareh Akbarzadeh, Patrick Gathura Ndungu

**Affiliations:** Energy, Sensors and Multifunctional Nanomaterials Research Group, Department of Chemical Sciences, Faculty of Science, University of Johannesburg, Doornfontein 2028, South Africa

**Keywords:** BiOCl, Ag_2_S, Bi_2_O_3_, desalination, molecular dynamic simulation

## Abstract

In this study, BiOCl based nanocomposites were used as photocatalytic membranes for a simulated study on water desalination in reverse osmosis membrane systems. Through molecular dynamic simulation, the molecular structure of BiOCl, BiOCl/Ag_2_S and BiOCl/Bi_2_O_3_ heterojunctions were designed and their electronic properties, mechanical properties, and membrane performance for water desalination were evaluated for the first time. The molecular structure was created, and a geometry optimization task was used to optimize it. Material Studio 2019 CASTEP was used for simulation of the electronic and mechanical properties and water desalination was performed by ReaxFF software under pressures between 0 and 250 MPa. The novel BiOCl based nanocomposites showed improved electronic and mechanical properties and, most importantly, improvements in salt rejection and water permeability as compared to well-known materials such as graphene and MoS_2_. BiOCl and BiOCl/Ag_2_S had a bandgap around two, which is the ideal bandgap for semiconductor photocatalysts. A salt rejection of 98% was achieved under an applied pressure of 10 MPa. Salt rejection was higher for BiOCl/Bi_2_O_3_, while water permeability was higher for BiOCl/Ag_2_S. The monolayer BiOCl was unstable under pressures higher than 50 MPa, but the mechanical stability of BiOCl/Ag_2_S increased twofold and increased fourfold for BiOCl/Bi_2_O_3_, which is even higher than MoS_2_. However, between the three nanocomposites, BiOCl/Ag_2_S was found to be the most ideal photocatalytic nanocomposite membrane.

## 1. Introduction

Membranes for water desalination are becoming more common for the separation of salt from saline water. The saline water passes through a membrane, which selectively passes water and separates the various salts. As the demand for water desalination increases, the need for developing more sustainable membranes increases. The currently available membranes suffer from low salt rejection/water permeability and in addition, have instability and fouling issues which reduce the lifetime of the membranes [1]. On the other hand, the new generation of photocatalytic membranes, which could improve salt rejection and increase water permeability, all while self-cleaning, is of interest [2]. Photocatalytic membranes integrate both physical and biological treatment; therefore, they are suitable for energy-efficient water treatment. Leong and his colleagues [3] reported a detailed review on TiO_2_ based photocatalytic membranes for water and wastewater treatment purposes. Yet these photocatalytic membranes often need additional support structures which are not stable under UV irradiation [4]. To overcome these drawbacks, selection of a photocatalyst that is stable and can work under visible light as well as perform water desalination with acceptable salt rejection and water permeability is necessary.

The practical application of photocatalysts for pollutant degradation depends on their efficiency, which is based on the effectiveness of semiconductors’ nanocomposites on absorbing visible/solar light [5]. The bandgap plays a vital role in semiconductor functionality, as the bandgap energy value, the values of the conduction band, and the valance band edges determine the photocatalytic activity of the photocatalyst. During the past few years, a lot of research has been directed towards the application of BiOCl as a visible light driven semiconductor which has a small bandgap [6]. However, absorbing the visible light is not enough to have an efficient photocatalyst as some photocatalysts have a very fast electron-hole recombination rate, which is a constraint on the effective degradation of pollutants. Designing a proper heterojunction of two or three semiconductors or nanocomposites could align the band edges and set the energy levels, which could reduce the recombination rate. Therefore, many researchers have studied constructing heterojunction semiconductors with BiOCl such as BiOCl/Bi_2_O_3_ [7], Bi_2_WO_6_/Ag_2_S [8], and BiOCl/Ag/BiVO_4_ [9]. Bi_2_O_3_ has been also proven to be a valuable photocatalyst but its usage is limited because of the costly fabrication process [10]. Bi_2_O_3_ has three different phases: α-, β-, and γ-Bi_2_O_3_. In many studies, α-Bi_2_O_3_ and β-Bi_2_O_3_ have been proven to have significant photocatalytic activity. In 2009, Seung Yong Chai and his colleagues [11] reported a visible light active BiOCl/Bi_2_O_3_ photocatalyst. Their results showed that BiOCl had a low photocatalytic efficiency under visible light irradiation; however, its heterojunction with Bi_2_O_3_ significantly improved the activity.

Silver sulphide (Ag_2_S) is another semiconductor that has shown promising photocatalytic activity [12]. However, Ag_2_S alone is not efficient for the degradation of pollutants, and in addition it has a relatively high cost [13].

In this study, the membrane performance of BiOCl, BiOCl/Bi_2_O_3_, and BiOCl/Ag_2_S were investigated and their electronic and mechanical properties were studied by molecular dynamics (MD) simulation. Material Studio 2019 with CASTEP and ReaxFF were used for simulation. In this study, BiOCl nanocomposites were introduced as a photocatalytic membrane and were simulated and investigated for their potential application in water desalination.

## 2. Computational Method Detail

### 2.1. Molecular Structure

A simulation box consisting of BiOCl-based nanocomposite (BiOCl, BiOCl/Ag_2_S, and BiOCl/Bi_2_O_3_) structure was created using Material Studio 2019. Separately, the simulation boxes for each component, including Bi_2_O_3_, BiOCl, and Ag_2_S, were produced for comparison. For the combination, two different trials were used to get the optimum weight percentage of each. Finally, BiOCl, BiOCl/Bi_2_O_3_ (70/30%), and BiOCl/Ag_2_S (70/30%) were selected and simulated as membranes. CASTEP (Cambridge serial total energy package), based on first-principles density functional theory (DFT), was used for calculation [14]. The exchange and correlation interactions were modelled using the generalized gradient approximation and the Perdew–Burke–Ernzerhof (PBE) function [15]. The cut-off for the kinetic energy of the electron wave function was 489.80 eV and the medium quality of the k-point sampling set was 6 × 6 × 6. In the geometrical optimization, all forces on atoms were converged to be less than 0.05 eV/Å. The maximum displacement was 0.002 Å with a maximum stress of 0.1 GPa. After creating the primitive cell, the lattice structures of BiOCl, BiOCl/Bi_2_O_3_, and BiOCl/Ag_2_S were investigated by applying the geometry optimization task, which minimized the total energy of each structure to an optimum one.

### 2.2. Water Desalination by Using ReaxFF Software

In this study, single layer membranes of BiOCl, BiOCl/Bi_2_O_3_, and BiOCl/Ag_2_S were designed and investigated for water desalination. A simulation box consisting of either a BiOCl, BiOCl/Bi_2_O_3_, or BiOCl/Ag_2_S sheet with a thickness of 11.2 Å was designed as a membrane. A total of 2000 water and salt molecules (1800 H_2_O, and 200 NaCl) were included, and external pressures to force the solution. A NaCl solution with 10% concentration was chosen as the feed solution to model the saline water. The membrane surface area was designed with a dimension of 100 × 70 Å^2^ in a simulation box with a dimension of 110 × 110 × 110 Å^3^ for a single-layer BiOCl based membrane. The NPT Berendsen simulation method was applied with a density of 0.2311 g/mL. The force field values for BiOCl and the elements in NaCl and water elements were selected from a software library [16]. The specific temperature inside the simulation box was selected as 323.15 K with a damping constant equal to 500 fs. The water permeability and salt rejection have been explained for all membranes under the applied pressure ranging between 0 and 250 MPa.

Water permeability and salt rejection of the suggested membrane nanocomposites (BiOCl, Bi_2_O_3_/BiOCl, and Ag_2_S/BiOCl) were evaluated under different applied pressure values ranging between 0 and 250 MPa, based on Equation (1):(1)$$Permeability=\frac{\left(membrane\;thikness\;\left(mm\right)\right)\left(amount\;of\;permeate\;\left(g\right)\right)}{\left(membrane\;surface\;area\;\left({cm}^{2}\right)\right)\left(time\;\left(s\right)\right)\left(differential\;pressure\;\left(bar\right)\right)}$$


The reverse salt rejection rates were calculated in terms of the Na^2+^ and Cl^−^ concentrations, respectively [17].
(2)$$R=\frac{N_{f}-N_{p}}{N_{f}}$$
where *R, N_p_, N_f_*, are the salt rejection rate, ion concentration (at t = 2.5 ns) in the permeate side, and ion concentration in the feed side, respectively.

### 2.3. Electronic and Mechanical Properties

The electronic and mechanical properties of the selected membrane materials, BiOCl, BiOCl/Bi_2_O_3_, and BiOCl/Ag_2_S, such as the bandgap, density of state (DOS), projection density of state (PDOS), Young’s modulus, bulk modulus, shear modulus, and Poisson ratio were investigated.

## 3. Results and Discussion

### 3.1. Molecular Structure

A molecular structure is a three-dimensional structure consisting of atoms arranged to make a molecule of interest that can be used to determine the polarity, reactivity, phase of matter, and biological activities of the compound [18]. Bi_2_O_3_, BiOCl, and Ag_2_S molecular structures were simulated based on the lattice constant values as indicated in Table 1.

The crystal lattice information was derived from the XRD data, tetragonal BiOCl (ref. code: 01-085-0861), monoclinic Ag_2_S (Ref code: 03-065-2356), and cubic Bi_2_O_3_ (Ref code: 01-083-3011). The generated crystal structures of BiOCl, BiOCl/Ag_2_S, and BiOCl/Bi_2_O_3_ are shown in Figure 1.

The geometry optimization task is the initial stage in the simulation process after creating a crystallographic structure of the materials based on their primary lattice parameters. The task was performed using Material Studio 2019 with 500 iterations. Figure 2 shows geometry optimization and energy convergence for the nanocomposite materials. As can be seen in Figure 2, for BiOCl, 7 iterations were enough to optimize the structure; for BiOCl/Bi_2_O_3_, 24 iterations were enough; and for BiOCl/Ag_2_S, 23 iterations were enough to optimize the structure.

A medium quality of the process under 2.0 × 10^−5^ eV/atom of energy with a maximum displacement equal to 0.002 Angstrom was selected. The process was used to minimize energy and organize the arrangement of atoms in a specific structure with the lowest energy configuration of the collection of atoms [19].

For BiOCl, the total energy of the initial structure before optimization was −2515.78 eV, and after optimization, the total energy was minimized to −2515.88 eV after 7 iterations. For the combinations of BiOCl/Bi_2_O_3_ and BiOCl/Ag_2_S, the total energy was minimized from −2513.23 to −2515.04 eV and from −33258.53 to −33259.87 eV, respectively.

### 3.2. Bandgap

The band structure was studied for BiOCl, BiOCl/Bi_2_O_3_, and BiOCl/Ag_2_S heterojunctions. Figure 3 shows the band structure of BiOCl, BiOCl/Bi_2_O_3_, and BiOCl/Ag_2_S. The calculated results showed that the BiOCl semiconductor has a bandgap (Eg) of 2.671 eV, which is similar to the value reported in a previous study [20]. The Ag_2_S showed a bandgap of 0.661 eV, which was reported by Vogel et al., 1994 [21]. Cubic Bi_2_O_3_ showed a very small bandgap almost equal to zero, which has been reported in a previous study [22]. The calculated bandgap value for the BiOCl/Bi_2_O_3_ and BiOCl/Ag_2_S composites were 0.762 and 2.398 eV, respectively, which could be used to predict the photocatalytic ability of the membrane. However, for Ag_2_S, the top of the valence band (TVB) and the bottom of the conduction band (BCB) occur in the G special *k*-point, while for BiOCl and BiOCl/Bi_2_O_3_ and BiOCl/Ag_2_S composites, TVB and BCB occur at G-point and Q-point, respectively. In addition, as shown in Table 2, BiOCl and BiOCl/Bi_2_O_3_ have a direct bandgap, which is similar to an earlier report by Das & Datta, 2020. Materials with direct band gaps are good absorbers for light photons and the sequence creation electron holes, whose lifetime is quite crucial for redox reactions in photocatalysis or band offset in optoelectronic device applications [23]. However, the band gaps of BiOCl and BiOCl/Ag_2_S composites are around two, which is the ideal bandgap for semiconductor photocatalysts [24].

### 3.3. Density of State (DOS) and Projected Density of State (PDOS)

DOS is another important analysis that also gives details about the electronic property of a material that measures the number of electron or hole states per unit volume at a given energy. DOS is also used to understand the type of semiconductors (n-type/p-type). Figure 4 show the DOS and PDOS for (a) *BiOCl*, (b) *BiOCl/Ag_2_S*, and (c) *BiOCl/Bi_2_O_3_*. For BiOCl (a,d), the conduction band is mainly constituted of O-s levels, while the valence band is constituted of Bi-p and O-s states. However, the gap appears between Bi-p and O-s orbitals. More careful analysis of the nature of the band edges reveals that the top valence band originates from the bismuth Bi-p orbital and the bottom conduction band originates from the oxygen O-s orbital.

For the BiOCl/Bi_2_O_3_ composite (c,f), the conduction band mainly constituted of Ag-d and S-p, while the valence band was mainly constituted of Ag-d. The gap appears between Bi-p and O-s. The top valence band originated from a bismuth, Bi-p orbital and the bottom conduction band originates from the oxygen O-s orbital. For BiOCl/Ag_2_S (e), the conduction band is mainly constituted of O-p and Bi-d, while the valence band also mainly constituted of O-p and Bi-d orbitals and the gap appears between Cl-s and Bi-d.

### 3.4. Mechanical Properties

#### 3.4.1. Bulk and Shear Modulus

The bulk modulus and the shear modulus were calculated by using the Reuss (*B_R_*) method [25] and the results are presented in Figure 5. For BiOCl, the bulk modulus value was approximately assumed to be equal to 137.7 GPa.

For the BiOCl/Bi_2_O_3_ composite, the bulk and shear moduli showed a significant increment, equal to 475 and 525 GPa, respectively. For BiOCl/Ag_2_S, the bulk and shear moduli equaled 345 and 360 GPa, respectively. This high increment in bulk and shear moduli values prove the high rigidity of the suggested composites to be useful in many applications with a promising result. BiOCl/Bi_2_O_3_ showed the highest bulk and shear moduli, indicating that the material is highly rigid and is not easily to deformed.

#### 3.4.2. Young’s Modulus

The graph in Figure 6 represents the Young’s modulus of elasticity for BiOCl, *BiOCl/Ag_2_S*, and *BiOCl/Bi_2_O_3_* composites as estimated by the universal forcefield. For BiOCl, the Young’s moduli was 231, 231, and 157 GPa in the x, y and z directions, respectively. For the *BiOCl/Bi_2_O_3_* composite, an increment in the Young’s modulus was observed as it reached 420 GPa in x and y directions, and 400 in the z direction. For BiOCl/Ag_2_S, the Young’s modulus values were increased to 320 GPa in the x and y directions, and to 302 GPa in the z direction. This shows that the mechanical response data obtained by Young’s modulus for the x, y, and z-axis of BiOCl/Bi_2_O_3_ and BiOCl/Ag_2_S has made the material more stable, while for pure BiOCl, the significant changes across the different axes could be observed (Figure 6). This indicates that creating composites could increase the Young’s modulus in all three directions and more interestingly, the same values in all directions.

#### 3.4.3. Poisson Ratio

The calculations of the Poisson ratios for BiOCl, BiOCl/Bi_2_O_3_, and BiOCl/Ag_2_S in three directions, as shown in Figure 7, has a positive value in all directions. For BiOCl, Exy = 0.32, Eyx = 0.32, and Ezx = 0.22, which is close to results reported in a previous study [26].

For the combination of BiOCl and Ag_2_S, the Poisson ratios decreased, which makes it easier to fracture and equal 0.2275 for all directions. For BiOCl/Bi_2_O_3_, the Poisson ratios increased to 1.082, 1.1233, and 0.0984 in Exy, Eyx, and Ezx, respectively. Meanwhile, the increase of the Poisson ratio for the combination of BiOCl and Bi_2_O_3_ makes it more resistant to the compression effect [27].

### 3.5. Water Desalination

#### Water Permeability and Salt Rejection

Figure 8 shows the simulating box consisting of a BiOCl photocatalytic membrane used in the simulation study for the water desalination process. Using Equation (1) [28], the amount of permeated water for the BiOCl, Bi_2_O_3_/BiOCl, and Ag_2_S/BiOCl photocatalytic membranes were calculated after 5 ns to investigate the performance of each BiOCl nanocomposite membrane in desalination under different applied pressure values ranging between 0 and 250 MPa.

At the beginning of the simulation work after 500 fs, H_2_O and NaCl molecules were added to the simulation box near the membrane. At 2 ns, 40% of H_2_O molecules were filtered for the combination of BiOCl/Ag_2_S under 250 MPa applied pressure. For BiOCl, 35% of H_2_O molecules were filtered under the same conditions. At 4 ns, 78% and 75% of H_2_O molecules were filtered for the composites of BiOCl/Ag_2_S and BiOCl/Bi_2_O_3_ under 250 MPa of applied pressure, respectively. For BiOCl at a high applied pressure of more than 50 MPa, the membrane was not stable, so we studied the permeability and rejection under lower applied pressure values ranging between 0 and 50 MPa for BiOCl.

However, at lower applied pressure values, the permeation rates were slow for all nanocomposites layers. Meanwhile, water molecules permeated approximately at a constant rate during simulation time because of the low effect of salt concentration in the condition studied here. Figure 9 shows the rate of water permeation through BiOCl, BiOCl/Ag_2_S, and BiOCl/Bi_2_O_3_ bilayers under different applied pressure values.

As shown in Figure 10, the salt rejection was investigated for BiOCl/Bi_2_O_3_, BiOCl/Ag_2_S, and BiOCl under external pressure values ranging between 0 and 250 MPa. Reverse salt rejection rates were calculated using Equation (2). In order to estimate salt passage, ions concentrations were calculated at the end of 2.5 ns since the salt rejection rate is time dependent. The salt rejection values were 90%, 92%, and 88% for the BiOCl/Bi_2_O_3_, BiOCl/Ag_2_S, and BiOCl bilayers, respectively, under 50 MPa of applied pressure after applying Equation (2) in the permeate and feed side at 2.5 ns. While under 250 MPa, the salt rejection percentage decreased to 70% and 68% for BiOCl/Ag_2_S and BiOCl/Bi_2_O_3_ respectively.

Comparing the performance of membranes for desalination reported in different literature is not easy, as the conditions used in each work is different. However, a table comparing this work with the results obtained from other work reported in the literature has been prepared (Table 3). In addition, the desalination performance of membranes that were studied under similar conditions in our previously published research [29,30] have been compared to BiOCl based membranes simulated in the present work. The results are summarized in Figure 11. Although the BiOCl membrane was a monolayer, when comparing it to bilayers of graphene and MoS_2_ based membranes, it had similar salt rejection and water permeability values to MoS_2_ under ultra-high operating pressures. In addition, it had better mechanical properties, since it shows higher shear and bulk modulus values, which indicate better membrane mechanical stability. The salt rejection of of BiOCl/Bi_2_O_3_ was higher, while water permeability was higher for BiOCl/Ag_2_S. The monolayer BiOCl was unstable under pressures higher than 50 MPa, but the mechanical stability of BiOCl/Ag_2_S increased by twofold and a fourfold increase was observed for the BiOCl/Bi_2_O_3_ when compared to the BiOCl, which is even higher than MoS_2_. The Young’s modulus was about 420 ± 50GPa and 310 ± 50 GPa for BiOCl/Bi_2_O_3_ and BiOCl/Ag_2_S, respectively, which is higher than the value of 270 ± 100 GPa as reported by Cao and colleagues for MoS_2_ [31].

These novel BiOCl/Bi_2_O_3_ and BiOCl/Ag_2_S nanocomposite membranes could significantly decrease the operational costs of desalination processes. The membranes have similar salt rejection and permeability values with higher stabilities, which are due to the better mechanical properties, and thus, they have competitive advantages over the performance of well-known membranes such as MoS_2_ and graphene.

## 4. Conclusions

BiOCl based nanocomposites materials were designed and simulated for water desalination for the first time. Their electronic structure and mechanical properties were studied and evaluated for water desalination. BiOCl/Ag_2_S and BiOCl/Bi_2_O_3_ composites showed improved mechanical and electronic properties compared to pure BiOCl. The band gaps of BiOCl and BiOCl/Ag_2_S composites were close to the ideal bandgap for semiconductor photocatalysts. It was predicted that the novel BiOCl based nanocomposites’ permeability would increase after creating a composite with Bi_2_O_3_ and Ag_2_S. A salt rejection of 98% was achieved under an applied pressure of 10 MPa. Salt rejection of BiOCl/Bi_2_O_3_ was higher, while the water permeability was higher for BiOCl/Ag_2_S. The monolayer BiOCl was unstable under pressures higher than 50 MPa but the mechanical stability of BiOCl/Ag_2_S increased by twofold and it was increased fourfold for BiOCl/Bi_2_O_3_, even higher than MoS_2_. Finally, between the three nanocomposites, BiOCl/Ag_2_S was found to be the most ideal nanocomposite membrane.

## Figures and Tables

**Figure 1 membranes-12-00505-f001:**
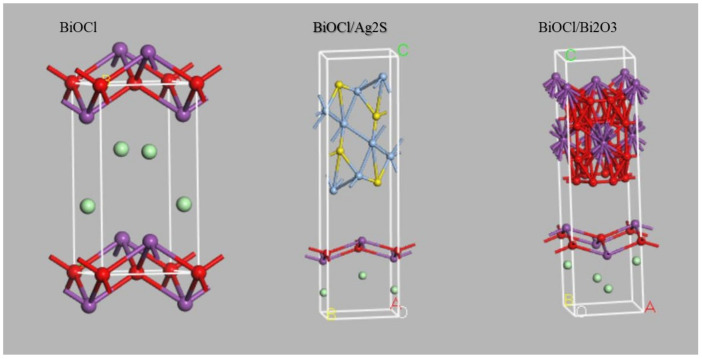
Molecular structure of: A BiOCl B BiOCl/Ag_2_S, and C BiOCl/Bi_2_O_3_.

**Figure 2 membranes-12-00505-f002:**
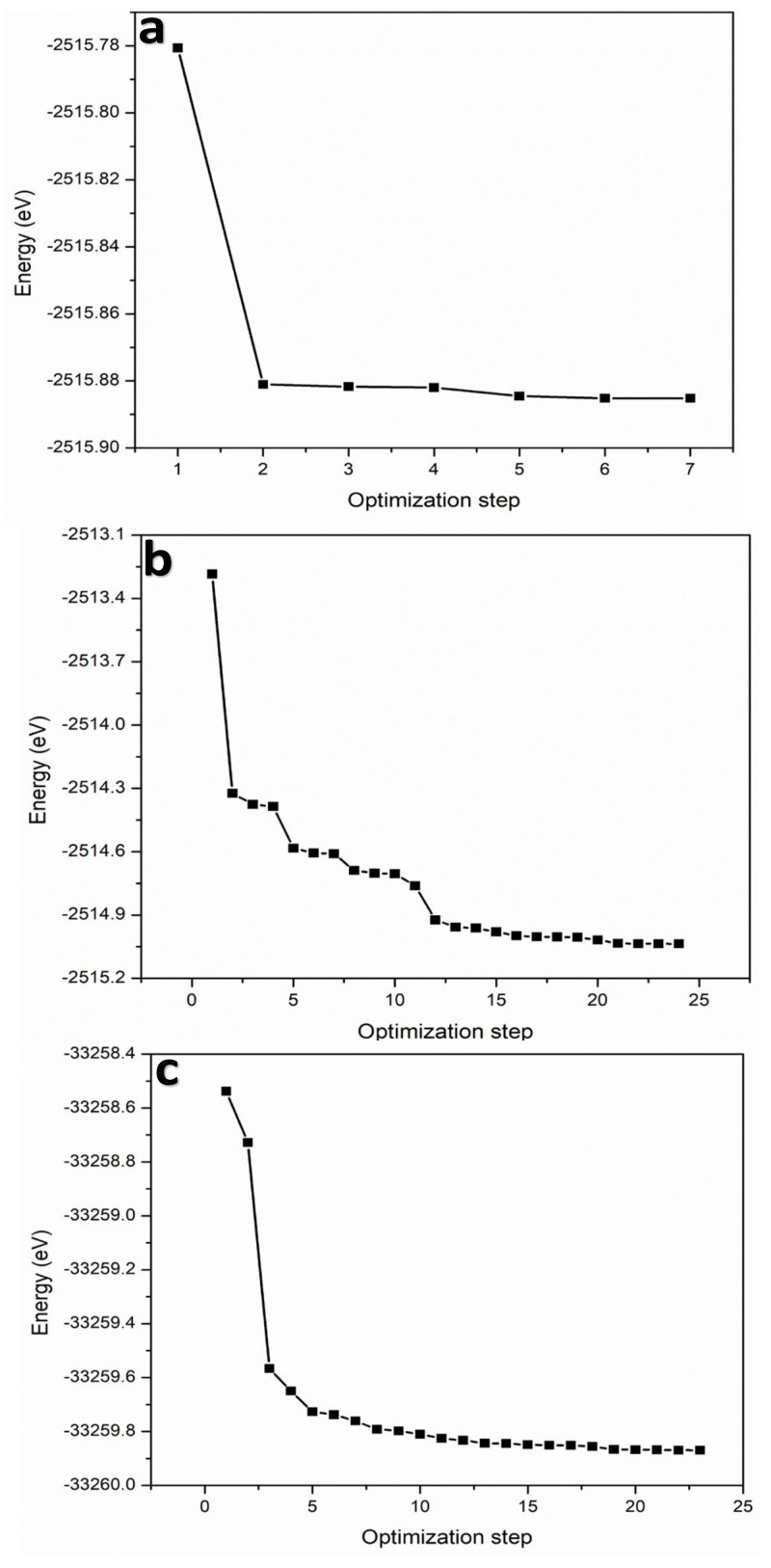
Atomistic energies for: (**a**) BiOCl, (**b**) BiOCl/Bi_2_O_3_, and (**c**) BiOCl/Ag_2_S.

**Figure 3 membranes-12-00505-f003:**
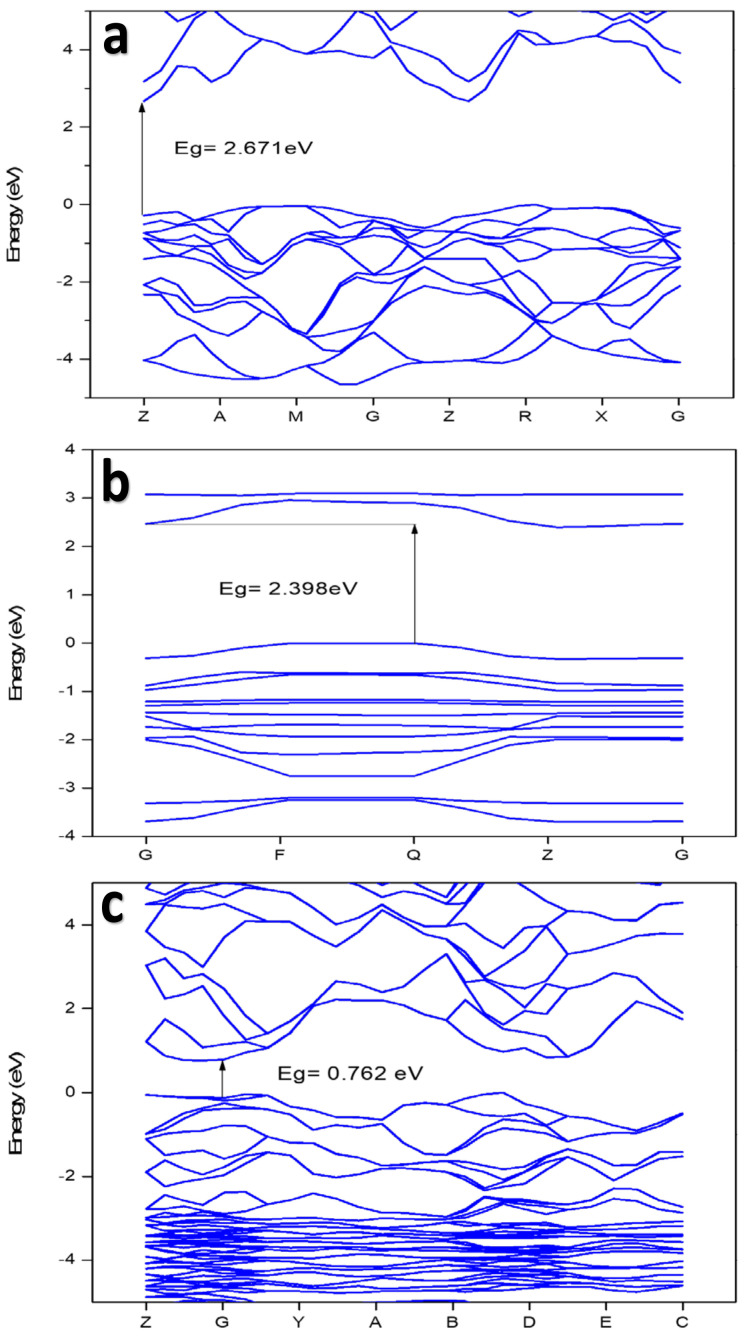
Calculated band structure of: (**a**) BiOCl, (**b**) BiOCl/Ag_2_S, and (**c**) BiOCl/Bi_2_O_3_.

**Figure 4 membranes-12-00505-f004:**
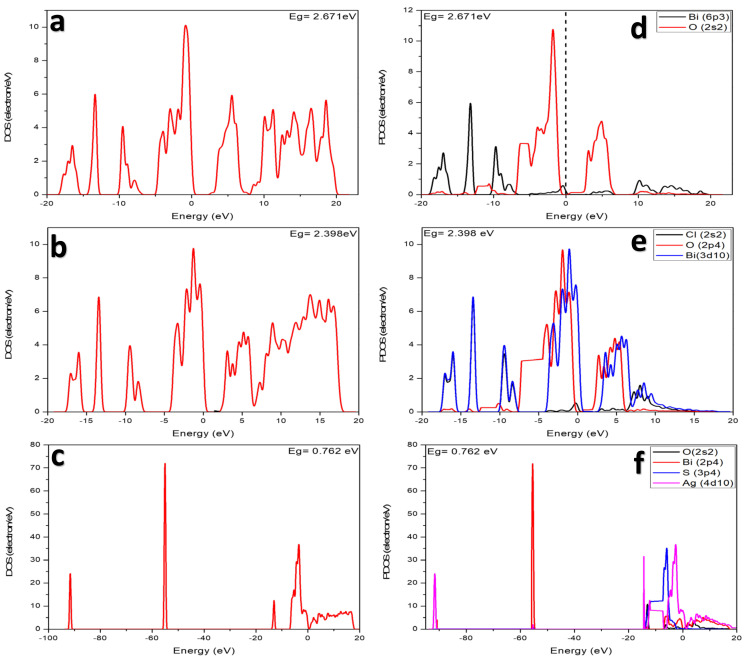
Calculated DOS of: (**a**) BiOCl, (**b**) BiOCl/Ag_2_S, (**c**) BiOCl/Bi_2_O_3_, and PDOS of: (**d**) BiOCl, (**e**) BiOCl/Ag_2_S, and (**f**) BiOCl/Bi_2_O_3_.

**Figure 5 membranes-12-00505-f005:**
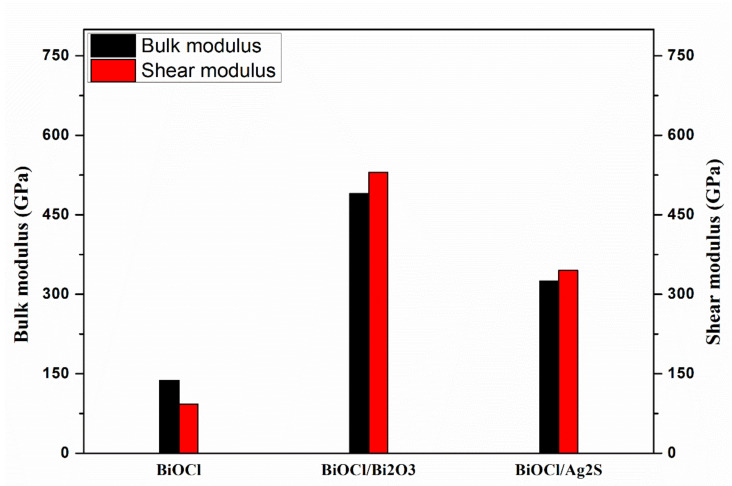
Bulk and shear modulus of BiOCl, Bi_2_O_3_/BiOCl, and BiOCl/Ag_2_S in three directions (x, y, and z).

**Figure 6 membranes-12-00505-f006:**
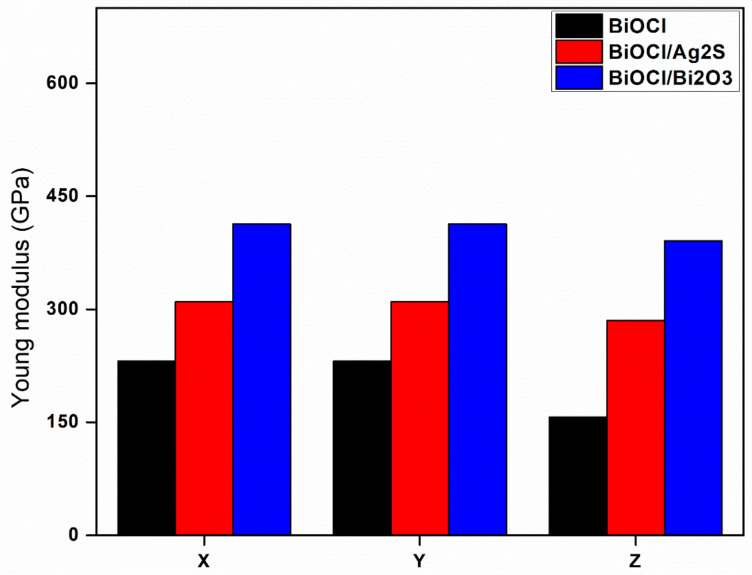
Young’s modulus of elasticity of BiOCl, BiOCl/Bi_2_O_3_, and BiOCl/Ag_2_S in three directions (x, y, and z).

**Figure 7 membranes-12-00505-f007:**
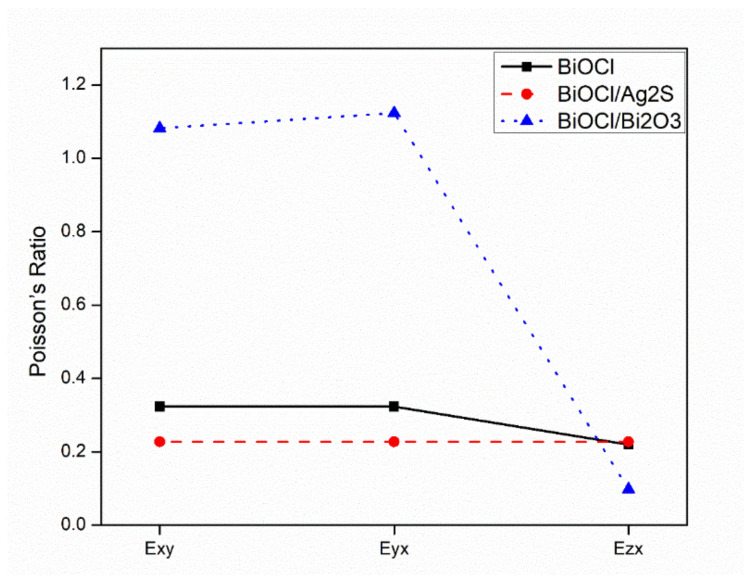
Poisson ratios of BiOCl, Bi_2_O_3_/BiOCl, and Ag_2_S/BiOCl in three directions (x, y, and z).

**Figure 8 membranes-12-00505-f008:**
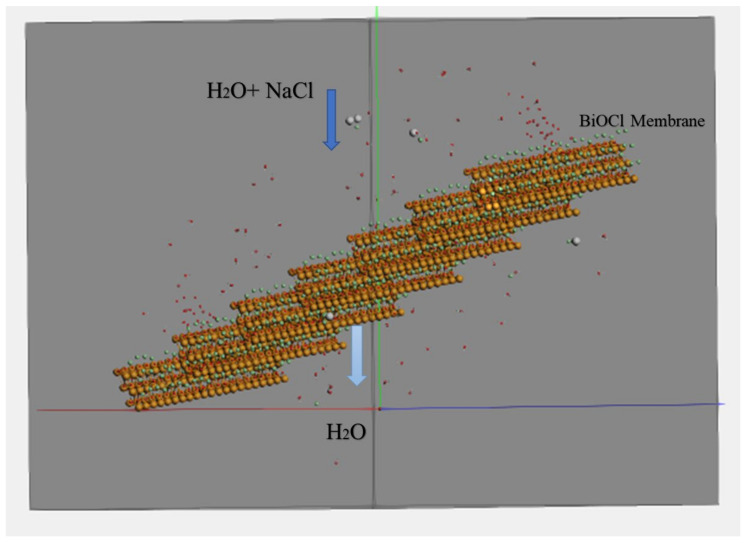
Simulation box consisting of BiOCl membrane molecules in orange, green and red, water molecules (H_2_O atoms) in red and white, NaCl molecules in white and green.

**Figure 9 membranes-12-00505-f009:**
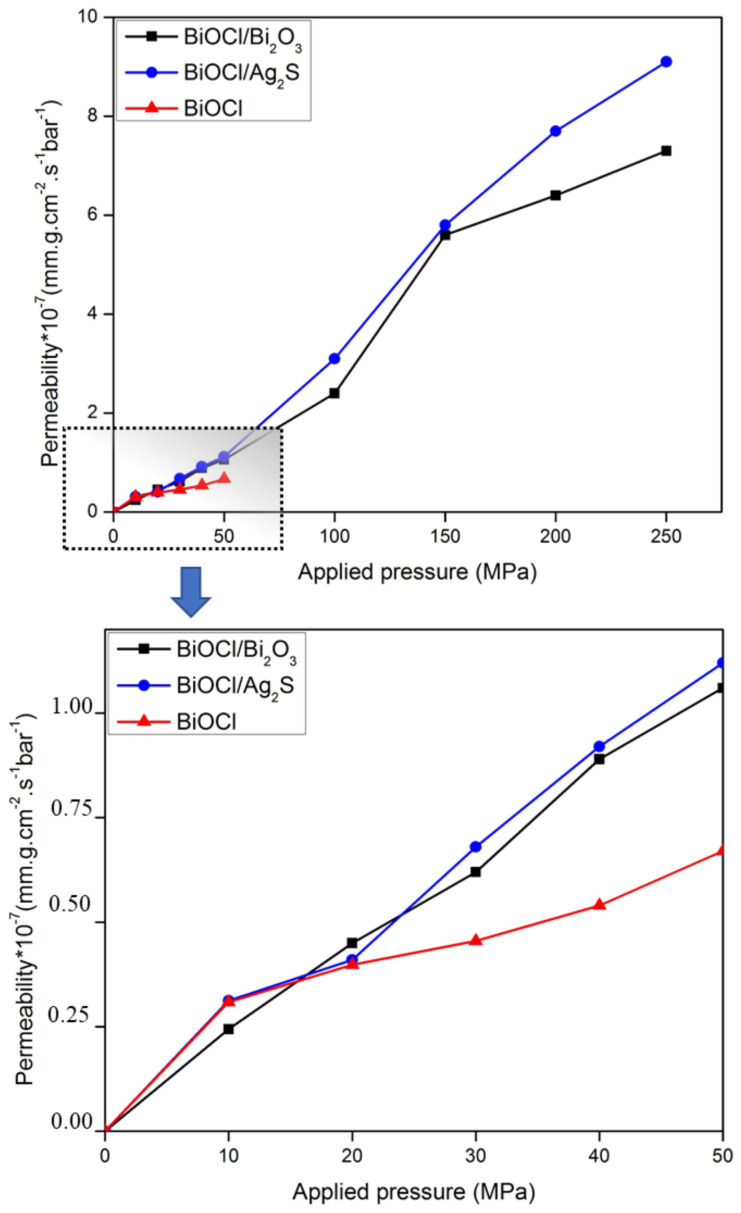
Water permeability rate of membranes under the suggested external pressure between 50 and 250 MPa.

**Figure 10 membranes-12-00505-f010:**
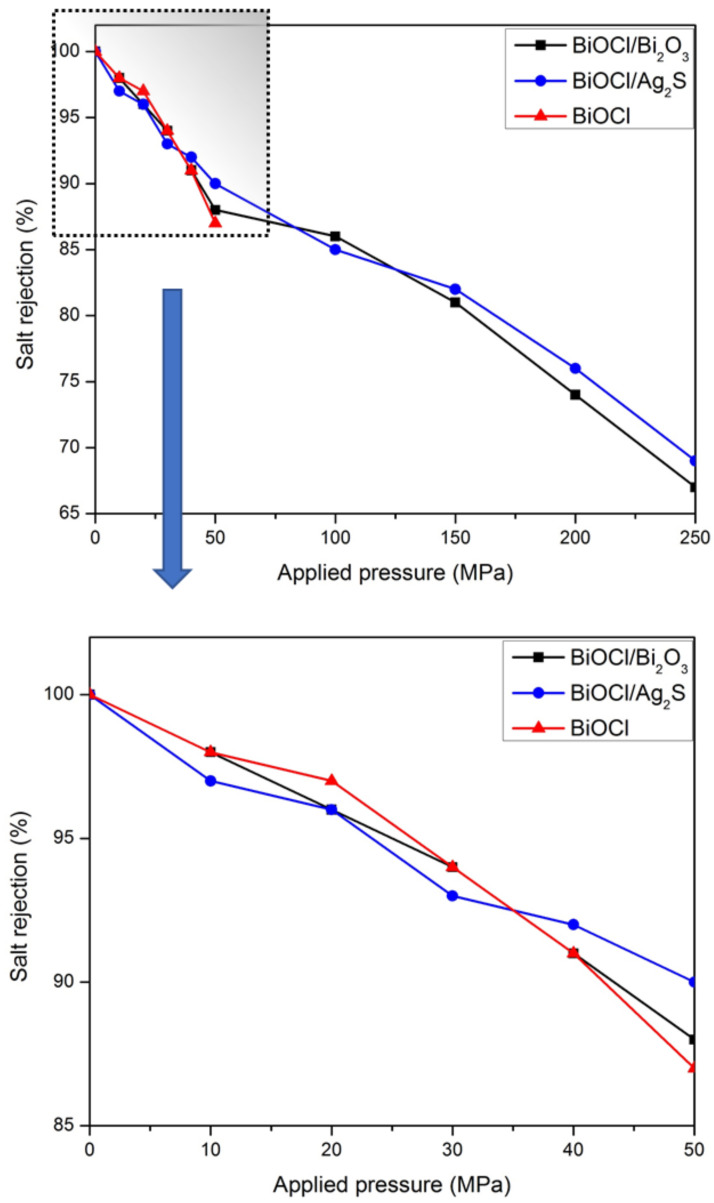
Salt rejection rate of membranes under the suggested external pressure between 50 and 25 MPa.

**Figure 11 membranes-12-00505-f011:**
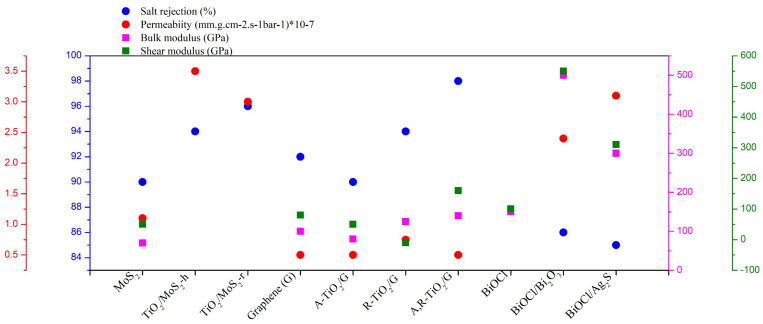
Salt rejection (blue), permeability (red), bulk modulus (magenta), and shear modulus (green) under 100 MPa pressure. Ps: MoS_2_, TiO_2_/MoS_2_-h, TiO_2_/MoS_2_-r [29], graphene, A-TiO_2_/graphene, R-TiO_2_/graphene, A-RTiO_2_/graphene [30].

**Table 1 membranes-12-00505-t001:** Lattice parameters for Bi_2_O_3_, BiOCl, and Ag_2_S.

Nanocomposite		Ag_2_S	Bi_2_O_3_	BiOCl
**Lattice constants (Å)**	a	4.23	5.6607	3.89
	b	6.91	5.6607	3.89
	c	7.87	5.6607	7.37
	α	90°	90°	90°
	β	99.58°	90°	90°
	γ	90°	90°	90°

**Table 2 membranes-12-00505-t002:** CB and VB for BiOCl, BiOCl/Bi_2_O_3_, and BiOCl/Ag_2_S.

Material	Bandgap (eV)	Band Type	CB (eV)	VB (eV)
BiOCl	2.671	Direct	2.241	0.430
BiOCl/Bi_2_O_3_	0.762	Direct	0.662	0.1
BiOCl/Ag_2_S	2.398	In-direct	2.101	0.297

**Table 3 membranes-12-00505-t003:** Comparison of membrane performance.

Membrane	Test Condition	Salt Rejection	Permeability or Flux	Young’s Modulus	Reference
MoS_2_	100 MPa pressure, 10 Å thickness, 100 K	99%	9.36 L cm^−2^ day^−1^ MPa^−1^	270 ± 100 GPa	[31]
Graphene	100 MPa, bilayer	85–100%	209 L m^−2^ h^−1^ bar^−1^	-	[32]
Polyamide	100 MPa, 1.28–5.40 nm thickness, 298.15 K	60–100%	2 K gm^−2^ h^−1^ × 10^3^	-	[33]
BiOCl	100 MPa pressure, 11.2 Å thickness, 323.15 K	-	-	200 ± 50 GPa	[This work]
BiOCl/Ag_2_S	100 MPa pressure, 11.2 Å thickness, 323.15K	85%	3 mm g cm^−2^ s^−1^ bar^−1^ × 10^−7^	310 ± 50 GPa	[This work]
BiOCl/Bi_2_O_3_	100 MPa pressure, 11.2 Å thickness, 323.15 K	86%	2 mm g cm^−2^ s^−1^ bar^−1^ × 10^−7^	420 ± 50 GPa	[This work]

## Data Availability

Data will be available upon request from the corresponding authors and based on University rules and regulation.

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
