# Peer review of "A Novel BiOCl Based Nanocomposite Membrane for Water Desalination"

_membranes, 2022, doi:10.3390/membranes12050505_

Round 1

Reviewer 1 Report

1- Please rephrase “This is the first time” to “Up to date, ….”.

2- The significance of the work should be highlighted clearly.

3- Standard errors are advised to be included in all the Figures.

4- All the equations are recommended to be transferred from the results and discussion to section 2 (Computational method detail).

5- The authors are strongly recommended to compare the outcome of the present work with the up to date references.

Author Response

1. Please rephrase “This is the first time” to “Up to date, ….”.

Response: This has been edited.

2. The significance of the work should be highlighted clearly.

Response: This has been added

3. Standard errors are advised to be included in all the Figures.

Response: This is a simulation work and at the beginning of the simulation to minimize the error the iteration runs for 500 which has been mentioned in the paper. 

4. All the equations are recommended to be transferred from the results and discussion to section 2 (Computational method detail).

Response: All equations are transferred to section 2 (materials and methods).

5. The authors are strongly recommended to compare the outcome of the present work with the up to date references.

Response: A table has been prepared and added for the comparison. A graph of performance comparison also has been prepared and added to the manuscript (Figure 11). We did compare this work with our previous published work as they are under similar condition.

Reviewer 2 Report

The paper entitled "A novel BiOCl based nanocomposite membrane for water desalination" is an interesting study about the evaluation of several composite membranes towards water desalination.

The manuscript is well-written and the scientific content is good.

I think the manuscript should be accepted in its present form.

Author Response

The paper entitled "A novel BiOCl based nanocomposite membrane for water desalination" is an interesting study about the evaluation of several composite membranes towards water desalination.

The manuscript is well-written and the scientific content is good.

I think the manuscript should be accepted in its present form.

Response: Thank you for your kind comments.

Reviewer 3 Report

The manuscript entitled “A novel BiOCl based nanocomposite membrane for water desalination” is well written. This can be considered worth publication after following revisions:

  1. Abstract: if abbreviation is not repeated in abstract then no need to mention it, such as RO and MD.
  2. Author should check the citation guidelines for this MDPI journal membranes, such as in text [1,2].
  3. Introduction: Author should provide the appropriate references for line # 24-28 and 36-40.
  4. Figure 1. Black color background should be removed for better understanding in print form.
  5. Figures 3,4,8 Should be replaced with better quality image.
  6. Author should provide a schematic diagram of setup for novel BiOCl based nanocomposite membrane for water desalination process.
  7. Conclusion section should be revised with core results of this study.
  8. The author can provide the comparison of this study with other related studies in tabular form for better understanding, as this novel membrane salt rejection showed 100-fold increase compared to materials such as graphene and MoS2.

Author Response

The manuscript entitled “A novel BiOCl based nanocomposite membrane for water desalination” is well written. This can be considered worth publication after following revisions:

Thank you for your kind comments. We have revised the manuscript based on your comments.

1. Abstract: if abbreviation is not repeated in abstract then no need to mention it, such as RO and MD.

Response: The abbreviations have been deleted from abstract.

2. Author should check the citation guidelines for this MDPI journal membranes, such as in text [1,2].

Response: The referencing has been modified based on MPDI journal membranes,

3. Introduction: Author should provide the appropriate references for line # 24-28 and 36-40.

Response: The appropriate references have been added for line #24-28 and 36-40.

4. Figure 1. Black color background should be removed for better understanding in print form.

Response: The image has been generated with light background

5. Figures 3,4,8 Should be replaced with better quality image.

Response: The Figures 3,4, 8 have been replaced with better quality images. Please note that in the revised version of paper the figures number have been changed.

6. Author should provide a schematic diagram of setup for novel BiOCl based nanocomposite membrane for water desalination process.

Response: The Schematic diagram has been extracted from the Material Studio software and have been added to the manuscript (Figure 8 in revised version).

7. Conclusion section should be revised with core results of this study.

Response: The conclusion has been revised and the core results have been presented.

8. The author can provide the comparison of this study with other related studies in tabular form for better understanding, as this novel membrane salt rejection showed 100-fold increase compared to materials such as graphene and MoS2.

Response: The manuscript has been revised and a comparison Table 3 and figure 11 have been added to compare the performance of BiOCl based membrane with MoS2 and graphene based membranes.

Round 2

Reviewer 3 Report

The authors have revised according to reviewer comments.